# Indications and Timing of Guided Growth Techniques for Pediatric Upper Extremity Deformities: A Literature Review

**DOI:** 10.3390/children10020195

**Published:** 2023-01-20

**Authors:** Mark F. Siemensma, Christiaan J.A. van Bergen, Eline M. van Es, Joost W. Colaris, Denise Eygendaal

**Affiliations:** 1Department of Orthopedics and Sports Medicine, Erasmus University Medical Center—Sophia Children’s Hospital, 3000 CA Rotterdam, The Netherlands; 2Department of Orthopaedic Surgery, Amphia Hospital, 4800 RK Breda, The Netherlands

**Keywords:** limb length discrepancy, alignment, growth correction, children, timing

## Abstract

Osseous deformities in children arise due to progressive angular growth or complete physeal arrest. Clinical and radiological alignment measurements help to provide an impression of the deformity, which can be corrected using guided growth techniques. However, little is known about timing and techniques for the upper extremity. Treatment options for deformity correction include monitoring of the deformity, (hemi-)epiphysiodesis, physeal bar resection, and correction osteotomy. Treatment is dependent on the extent and location of the deformity, physeal involvement, presence of a physeal bar, patient age, and predicted length inequality at skeletal maturity. An accurate estimation of the projected limb or bone length inequality is crucial for optimal timing of the intervention. The Paley multiplier method remains the most accurate and simple method for calculating limb growth. While the multiplier method is accurate for calculating growth prior to the growth spurt, measuring peak height velocity (PHV) is superior to chronological age after the onset of the growth spurt. PHV is closely related to skeletal age in children. The Sauvegrain method of skeletal age assessment using elbow radiographs is possibly a simpler and more reliable method than the method by Greulich and Pyle using hand radiographs. PHV-derived multipliers need to be developed for the Sauvegrain method for a more accurate calculation of limb growth during the growth spurt. This paper provides a review of the current literature on the clinical and radiological evaluation of normal upper extremity alignment and aims to provide state-of-the-art directions on deformity evaluation, treatment options, and optimal timing of these options during growth.

## 1. Introduction

In contrast with adults, children have the unique capability to correct bone deformities by growth. Most deformities have a traumatic origin. Traumatic injury can occur at the level of the epiphysis, the physis, the metaphysis, or diaphysis [1]. Most correction can be expected in younger children in deformities near the most active growth plate and in the direction of the dominant movement. Most often, growth behaves like a friend, allowing the deformity to correct itself naturally during extensive follow-up. Natural correction follows two principles. The first principle is formed by the Hueter–Volkmann law, in which a degree of pressure on the convex side increases periosteal bone formation and relatively inhibits longitudinal growth [2]. Conversely, bone formation on the concave side is stimulated, increasing bone formation and resulting in a relative increase of longitudinal growth. In the second principle, Wolff’s Law addresses the ability of the bone and joint surfaces to remodel according to local mechanical loads [3].

Sometimes growth behaves less favorably, leading to severe malunions, joint instability, joint incongruence, impairment of movement, and eventually to early posttraumatic arthritis. Especially fractures through or nearby the physis are renowned for problems during growth, such as premature closure caused by bone bars, resulting in asymmetrical growth or even growth arrest. It is, therefore, of great importance to closely monitor these cases and, if needed, intervene surgically to correct the deformity.

While in the lower extremity, guided growth by epiphysiodesis or hemi-epiphysiodesis is described extensively, little is known about the techniques, timing, and principles in the upper extremity. The purpose of this paper is to provide a review of the current literature concerning the clinical and radiological evaluation of normal upper extremity alignment and the causes and assessment of deformities in the upper extremity. Moreover, it aims to provide state-of-the-art directions on deformity evaluation and monitoring, different treatment options, and the timing of these options during growth.

## 2. Methods

For this narrative review, the Medline Ovid and Embase databases were searched for peer-reviewed studies in English until 30 November 2022. The search was divided into subcategories regarding each section using the following keywords and synonyms. Clinical and radiological evaluation: alignment, carrying angle, Baumann angle, Hafner method, cubitus varus, physical examination, radiograph. Causes of deformity: epiphyseal plate, (Salter–Harris) fracture, ischemia, repetitive stress, physeal plate, Madelung deformity. Treatment options: eight-plate, tension band, transphyseal screw, physeal bar resection Timing of intervention: skeletal age, growth chart, multiplier method, guided growth. A total of 716 articles were screened by title and abstract for relevance, twenty-two articles were additionally included by snowballing, and 83 articles were included in the final synthesis. All articles were screened by two independent reviewers. Single case reports and studies based on expert opinions were excluded from the synthesis. Because symptomatic deformities of the upper extremity occur primarily at the distal parts of the humerus and forearm with relation to the elbow and wrist, other upper extremity parts are deemed outside of the scope of this review. All patients and parents gave written informed consent for the publication of the anonymized images used in this review.

## 3. Clinical Evaluation

Standard physical examination is a quick and reliable method for the assessment of gross deformities of the upper limb [4]. Comparison to the uninjured side is mandatory in posttraumatic deformities. The examination should include the assessment of alignment in three directions, joint effusion, active and passive range of motion, evaluation of stability and points of tenderness, and finally, neurovascular assessment.

Because symptomatic deformities of the upper extremity occur primarily at the distal part of the humerus and in the forearm, range of motion (ROM) of the elbow and wrist is important to assess. Pediatric ROM of both the elbow and the wrist is slightly increased compared to adults, caused by hyperlaxity of the joints in childhood (Table 1). Despite the slight differences, a separate set of normal values is not defined for children in most hospitals, and the adult ROM-values are often used during clinical evaluation.

### 3.1. Elbow

During visual inspection, the patient is placed in front of the examiner, standing in the anatomical position with both elbows fully extended (Figure 1). The carrying angle is measured best in a standing patient with arms in the anatomical position, with arms fully extended and wrists fully supinated. The carrying angle is the angle deviated from the line parallel to the humerus and the forearm. The carrying angle is usually greater in women, with an average of 15–20 degrees. In men, the carrying angle is, on average, 10–15 degrees [5,6,7]. Therefore, the carrying angle is compared best with the contralateral side. A physiological change in the carrying angle from valgus to varus can be observed as the patient flexes the elbow and supinates the forearm [8]. Assessment of the carrying angle is therefore performed in the same amount of flexion and rotation of both arms to optimize adequate comparison. In most female patients and children, a slight hyperextension of the elbow of 0 to 10 degrees is physiological [9]. Deformities in the sagittal plane are measured with the humerus in 90 degrees anteflexion and by flexing and extending the elbow with the forearm in a supinated position.

Forearm rotation is measured by having the patient place the arms parallel to the side of the patient with the elbows in 90 degrees flexion. A pen in the child’s fist improves rotatory measurements.

### 3.2. Wrist

There are multiple ways of measuring wrist dorsal flexion. One quick way to assess for gross deformities is to have the patient put their palms together with their fingers pointing upwards and their elbows kept horizontally. Wrist palmar flexion can be performed similarly by putting both dorsal sides of the hand together with the fingers pointing downwards. Angular deviation in the coronal plane is measured by the radial and ulnar deviation of the wrist. It is measured best by having the patient place the supinated forearm on a flat surface with the palm of the hand lying flat. A small reference line is then drawn on the dorsum of the hand along the third metacarpus.

## 4. Radiological Evaluation

### 4.1. Humerus

Different radiographic angles have been described for the assessment of humeral alignment. The Baumann’s angle is measured on an anteroposterior (AP) radiograph with the elbow in extension. It is formed by the angle between the long axis of the humeral shaft and a straight line through the epiphyseal plate of the capitellum or the lateral condylar physis (Figure 2A). There is a considerable variation in individuals, ranging from 64 to 82 degrees [10,11]. Therefore, the Baumann’s angle is best compared to the contralateral side, where a difference >5 degrees is deemed abnormal.

The lateral capitellohumeral angle (LCHA) is measured on a lateral radiograph as the angle between the line along the anterior surface of the humerus and a line along the open capitellar physis (Figure 2B). The LCHA has a smaller normal range from 45 to 57 degrees and does not vary by age, side, or sex [12] (Figure 2B).

The lateral anterior humeral line (AHL) or capitellohumeral line is a line drawn along the anterior surface of the humerus, which should pass through the middle third of the capitellum on a lateral view (Figure 2C).

### 4.2. Radius

Radial height is measured on posteroanterior (PA) views as the distance between two parallel lines: one perpendicular to the long axis of the radius along the ulnar aspect of the articular surface and the other one at the tip of the radial styloid (Figure 3A). A normal adult radial height is 8 to 14 mm [13]; however, the values range in the literature, and those for children are unknown.

Volar tilt is measured on the lateral view as the angle between a line drawn perpendicular to the long axis of the radius and a tangent line along the slope of the articular surface of the radius (Figure 3B). A normal volar tilt ranges between 10 and 25 degrees [14,15].

Radial inclination is measured on the PA view as the angle between a line perpendicular to the long axis of the radius at the level of the radial styloid tip and a line along the articular surface of the distal radius (Figure 3C). A normal radial inclination ranges between 15 and 25 degrees [14,16].

### 4.3. Ulna

Ulnar variance, or Hulten variance, can be positive, negative, or neutral. In positive ulnar variance, the ulna is longer than the radius. Conversely, in negative ulnar variance, the radius is longer than the ulna. In neutral ulnar variance, both the articular surfaces of the radius and the ulna are at the same height. Ulnar variance is independent of the length of the ulnar styloid process, which may also vary.

There are multiple ways of measuring the ulnar variance. In the method of perpendiculars, a line is drawn perpendicular to the longitudinal axis of the radius and through the most distal ulnar part of the radius. The distance between the adjacent distal cortical rim of the ulna relative to this line is then measured as the variance [17] (Figure 4A). In the method as described by Hafner et al. [18], a line is drawn perpendicular to the longitudinal axis of the ulna, touching the most proximal prominent point of the ulnar metaphysis on the radial side. Secondly, a line is drawn on the radius perpendicular to its longitudinal axis touching the most proximal point of the radial metaphysis on the ulnar side. Ulnar variance is then defined as the distance between these lines. In the literature, these distances are referred to as “Proximal–PRoximal distance [17]. Conversely, the variance can be measured using the distance of the most distal points of the radial and ulnar metaphysis. This method is referred to as “Distal–DIstal” distance. (Figure 4B).

Kox et al. investigated the difference between the above-stated methods for measuring ulnar variance in a group of 350 healthy children and adolescents. It was found that the Hafner method was the preferred method for children with unfused growth plates or those younger than 13 years, and the adapted perpendicular method was recommended in children with fused growth plates or those 14 years and older [17].

Ulnar variance changes with wrist position and during clenching of the wrist. It is more positive during pronation and becomes more negative during supination. In addition, a clenched fist results in a relatively more ulna plus compared with a neutral grip. Therefore, obtaining only a PA-view with a neutral grip may underestimate maximal variance, and obtaining clenched fist view radiographs can be a useful addition [15].

## 5. Causes of Upper Extremity Deformity

Osseous deformities can arise at different levels of the bone, dependent on the pathophysiology. Deformities occurring at the level of the diaphysis and the metaphysis are often the result of a malunited fracture [1]. Deformities at the level of the physis have a broader spectrum of causes. They can be congenital, developmental, or acquired as the result of an infection, arthritis, compartment syndrome, avascular necrosis, or trauma, with the latter being the most common cause [18,19].

In up to 10% of physeal fractures, some form of growth disturbance occurs [20,21,22]. The main traumatic factors contributing to the growth arrest of the physeal plate are crush fractures from high-energy injury or repetitive stress (i.e., Salter–Harris type V) and physeal injuries crossing the germinal layer (i.e., Salter–Harris type III and IV) [21]. Traumatic growth disturbance may cause slower, asymmetrical, or arrested growth. These growth disturbances are often the result of an incorrect or overstimulated fracture repair. During fracture healing, when blood vessels reach the hypertrophic zone of the physis, ossification is stimulated, and a physiologic bridge of sclerotic bone forms eccentrically between the epiphyseal ossification center and the metaphyseal bone [20]. This effectively replaces a segment of the physis and the zone of Ranvier [23]. The effects of this bony bridge vary with its location and size but will result in either a complete or a partial growth disturbance. A large central bar will slow down or completely arrest the growth of the entire physis, creating a short bone, which in term may lead to limb length inequality or joint congruity if the bone of a pair is affected in the case of the radius and ulna. When the bar is eccentrically formed within the physis, growth stops at that point but continues in the rest of the physis. This results in a progressive angular deformity [24].

### 5.1. Humerus

Cubitus varus is most often seen as late sequela after a distal humerus fracture (Figure 1B). The current stance in the literature is that it is caused by malunion of a humeral fracture rather than a growth arrest. The most common type of distal humerus fracture in children is the supracondylar fracture [25]. They are classified using the Gartland type classification, ranging from type I to type III, depending on the amount of posterior displacement of the capitellum and the intactness of the posterior humeral cortex [26]. Cubitus varus, however, results from displacement or comminution in the coronal plane. These injuries are often overlooked or difficult to judge on standard radiographs. Therefore, in type II and type III fractures, an oblique view may be helpful in identifying minimally displaced fractures [27]. Rotational malalignment can be difficult to assess radiographically. A high index of suspicion for rotational malalignment is required in cases of posteromedial displacement. These cases may also lead to a higher Baumann angle and hence combined cubitus varus deformity [28]. If missed or left untreated, the malunion leads to a progressive angular deformity in the coronal plane. The result at patient presentation is often a painless varus deformity evident at visual inspection that may not always be accompanied by limitations in ROM [25]. Although diagnosis is usually based on clinical evaluation alone, measuring the radiological Baumann angle compared to the contralateral side may give a more accurate measure of the extent of the deformity.

### 5.2. Radius

Proximal radial fractures represent up to 10% of all pediatric elbow fractures [29,30]. The mechanism of injury is usually a fall on an outstretched hand, combined with a compressive valgus force across the elbow joint. Despite the occurrence of these fractures around the growth plate, premature physeal closure occurs in about 1.5% of patients [30]. Growth has more impact, however, in congenital radial head dislocations (Figure 5). Although rare in absolute numbers, it is the most common congenital elbow abnormality, accounting for up to 10% [31]. Dislocations occur bilaterally in most cases. Around 70% of dislocations occur posteriorly, followed by anterior and lateral dislocations, occurring around 15% each [31]. With frequent dislocations, the normal anatomical relation of the radial head with the capitellum and the proximal radioulnar joint (PRUJ) during growth may be lost. Without the pressure of the radial head onto the capitellum during growth, a malformation of the radial head with loss of concavity occurs, making reduction in longstanding cases impossible [31]. Patients are generally presented with a painless mass at visual inspection or palpation. Elbow flexion may be slightly decreased in the case of an anterior dislocation, and extension may be slightly decreased in the case of a posterior dislocation. Additionally, DRUJ alignment may be lost, resulting in decreased ROM during pronation and supination [31]. A lateral elbow radiograph is often sufficient to diagnose this condition. Herein, the extent of radial head deformation is a reliable guideline in the decision of whether to operate on a patient [32]. If the radial head is more dome-shaped and has lost all its concavity, surgery tends to be unsuccessful.

The distal radius is the most common site of physeal injury of the upper extremity, accounting for 30% to 39% of all physeal injuries [33]. The incidence of growth disturbances in the forearm caused by distal physeal injuries has been reported at a rate of up to 28% [18]. Despite the generally good outcomes of distal radius fractures, the incidence of a premature complete distal radius growth arrest is up to 7% [34,35,36]. Risk factors for developing premature physeal arrest are repeated forceful manipulation during reduction, multiple reduction attempts, and late reduction [34]. A posttraumatic radial physeal arrest can result in ulnar overgrowth, otherwise known as positive ulnar variance or ulna-plus. This occurs when the level of the ulna is >2.5 mm beyond the radius margin of the distal radioulnar joint (DRUJ). Gross deformity develops if the discrepancy between the radial and the ulnar length is more than 4 mm [34]. Despite the presence of gross deformity, functional problems do not always occur, and therefore, clinical presentation can be variable [18]. However, the majority of patients report significant impairment, most commonly by activity-related pain and loss of pronation–supination. Some asymptomatic patients in which radiographic signs of physeal arrest and positive ulnar variance are seen may opt for early surgery to prevent progressive deformity [36].

Chronic repetitive stress injuries of the distal wrist are increasingly being mentioned as a distinct diagnosis. This type of injury has a high incidence in competitive gymnasts. It is, therefore, also known as the ‘gymnast wrist’. As the result of repetitive axial loading with microtraumata of both the distal radial and ulnar physis, premature closure of the physis can occur, mimicking a Salter–Harris type V injury [33]. In a study by DiFiori et al. in which fifty-nine gymnasts were examined, 51% had radiological findings of stress injury to the distal radial physis, and 7% had distinct widening of the growth plate. In addition, wrist pain was significantly related to the grade of radiographic injury. Prolonged repetitive stress on the distal radial physis can even lead to complete physeal arrest [37]. Radiologic criteria for the diagnosis of stress injuries in the physis of the distal radius include widening of the growth plate, especially on the volar and radial side, cystic changes of the metaphyseal aspect of the growth plate, a beaked distal volar and radial physis, and haziness within the growth plate [38]. These criteria are named in multiple reports; however, a comprehensive guideline for the classification of these injuries is lacking [38,39].

Madelung deformity is a rare congenital arm condition that affects the growth plate of the distal radius. The lagging growth of the distal radius results in a radioulnar and radiocarpal misalignment. The progressive growth disturbance may eventually lead to a three-dimensional wrist deformity [40]. Madelung deformity is usually diagnosed between the ages of 6 and 13 years [41]. In children with Madelung deformity, additional ulnar radiological measurements are indicated. Farr et al. [40] stated that in addition to ulnar variance, a lunate subsidence (LS) >4 mm and a palmar carpal displacement (PCD) >20 mm were radiographic criteria for undergoing an ulnar shortening osteotomy. They measured PCD on a lateral radiograph as the distance between the longitudinal ulna axis and the most volar aspect of the lunate (Figure 6A). LS was measured on a PA radiograph as the distance between a perpendicular line to the longitudinal ulna axis and the most proximal point of the lunate. (Figure 6B). Symptoms of Madelung deformity can range from wrist pain to decreased function. Most commonly, patients experience a limited range of motion in the wrist and continuous or post-activity wrist pain.

### 5.3. Ulna

In the proximal part of the ulna, physeal fractures of the olecranon account for 4% of all pediatric elbow fractures [42]. They usually occur as a result of a fall onto an outstretched hand with the elbow in flexion. Nondisplaced fractures respond well to conservative treatment, and growth disturbances are rare. Growth disturbances in the distal ulna, caused by physeal injuries, however, have been reported at a rate of up to 50% [18]. The higher percentage may be explained by the higher force required to overcome the cushioning effect of the cartilage between the ulna and the proximal carpal row and the dissipation of impact forces through the triangulate fibrocartilage complex [43]. A shortened distal ulna results more commonly from any of the surgical procedures that involve resection of the distal ulna secondary to prior wrist trauma or correction of Madelung deformity.

### 5.4. Combined Radioulnar

During forearm rotation, the relation in the distance between the radius and ulna changes dynamically. Angular deformities in either bone can further increase or decrease the distance between both bones during rotation [44,45,46]. During pronation, the radius crosses the ulna, and their respective distance decreases. Radial bowing and radius malunions with the deformity pointed towards the ulnar side may cause a pronation deficit by a collision of the radius and ulna [45,46]. Conversely, during supination, an increase in distance between both bones is seen. The radius and ulna are interconnected by the central band. This ligament allows for the dissipation of forces from one bone to the other but can also pose problems in the case of osseous deformity. In radius malunions directed away from the ulna, the rigid central band length can impair further rotation and cause a supination deficit [45,46].

## 6. Treatment Options

In contrast to the lower extremity, where even minor limb length differences can lead to symptoms, minor differences in length in the upper extremity pose a lesser problem. In general, expected length differences of less than 5 cm in the humerus are generally treated conservatively. If the bones of the distal radius and ulna are affected, the margins are smaller. Radioulnar variance greater than 4 mm is considered a gross deformity [34]. Any physeal arrest in either the radius or the ulna can therefore be a good indication for surgical intervention. In general, treatment options for physeal arrest include observation, (temporary-) epiphysiodesis or hemiepiphysiodesis, physeal bar resection, and corrective osteotomy.

### 6.1. Observation

Growth arrest, angular deformities, and consequentially altered joint mechanics may develop up to 2 years post-injury [47]. Secondary to the injury, damaged cartilage tissue within the physis is often replaced by unwanted bony tissue, forming a bony bar or bony bridge. If the fracture is aligned correctly with or without reduction, physicians may choose for casting to ensure anatomic alignment and to prevent displacement accompanied by close radiological follow-up for observation of bony bar formation [48]. If a bar appears to involve the entire physis and the predicted length inequality or angular deformity at skeletal maturity is acceptable, observation may be the best option. Because growth often naturally corrects the deformity, another consideration can be to initially observe the deformity until skeletal maturity and to plan a correction osteotomy to correct the deformity if needed.

### 6.2. Hemiepiphysiodesis

In (progressive) angular deformities, hemiepiphysiodesis can be performed by tethering the proximal and distal physeal parts together. This results in a temporary halt of the growth at one side while the other side can catch up, correcting the deformity by growth. Growth plates can be tethered together using metal clips over the physis, by drilling screws through the physis, or by connecting the proximal and distal part of the physis together by non-resorbable filament, Kirschner wires, or a nonlocking plate that acts as a tension band. Due to its reversibility, this technique is safer and more predictable than a classic permanent epiphysiodesis. In addition, the exact timing of the intervention is of lesser importance because the implant is removed when the desired correction is achieved. The required second surgery to remove the implant, however, is a considerable disadvantage compared to permanent epiphysiodesis [49].

In the lower limb, modulation of growth by tethering part of the growth plate using tension-band plates (TBPs) or eight-plates is an established technique. The literature shows high efficacy and low complications with success rates for correction up to 93% [50,51]. Despite the high efficacy of the technique, no cohort studies of sufficient size have been published using TBPs in the upper extremity. A rebound phenomenon after using tension band hemiepiphysiodesis is known to occur. This happens when the growth of the inhibited side of the physis exceeds that of the contralateral side due to transient overstimulation after tension band removal [52]. To compensate for this, a slight overcorrection can be aimed for. A high correction rate is a significant risk factor for developing overcorrection. This is found to be a direct indicator of physeal activity, wherein a higher rate of correction is indicative of a larger residual growth plate activity [52]. Younger age at initial surgery and implant removal may also pose a risk factor [53,54]. The younger the patient is at the initial procedure, the higher the growth plate activity, leading to a more rapid correction and concomitant longer time between plate removal and skeletal maturity. Most studies advocate delaying temporary hemiepiphysiodesis until 8–10 years for the lower extremities due to the occurrence of rebound or concerns about causing permanent physeal damage [55]. Despite the lower growth rates of the upper extremity compared to the lower extremity, clinicians should monitor patients closely after tension band hemiepiphysiodesis for rebound phenomena, especially in younger patients.

An alternative technique to tether the growth plate is to use transphyseal screws (Figure 7). This technique has a faster correction rate than the tension band principle [56,57]. Hence, this technique may better serve patients that are near skeletal maturity. Soldado et al. [58] used transphyseal crossed cannulated screws (Metaizeau technique) to correct cubitus varus deformities in five very young children. The children had a mean age of 3 years and 7 months and were followed over a mean period of 3 years and 10 months. No correction was observed in all cases. The authors postulated that the ineffectiveness may be explained by the modest growth capacity of the distal humeral physis and because most growth occurs during the pubertal growth spurt, while their follow-up finished before any of their patients reached that stage. Dai et al. [59] studied temporary hemiepiphysiodesis in a total of 135 physes in 66 children with a mean age of 4.69 years old (ranging from 1 to 10 years). In a mean deformity correction period of 13.26 months, 94.06% of the angular deformities were corrected. Thus, posing temporary hemiepiphysiodesis using the Metaizeau technique is an effective method for correcting angular deformities in younger children. A probable reason why the deformity correction for young children in the lower extremity is more successful than in the distal humerus is the difference in axial growth speeds and the percentage of contribution of the physes with regard to the total limb growth. Only 20% of growth takes place in the distal humerus, accounting for a mean of 0.26 cm per year. Conversely, in the distal femur and proximal tibia these percentages are 70% and 60%, respectively, which corresponds to 1.2 cm and 0.9 cm per year [60,61,62].

### 6.3. Complete Epiphysiodesis

In complete epiphysiodesis, the physis is completely removed or temporarily tethered across the entire width. This procedure is performed to prevent overgrowth. Surgical options range from percutaneous techniques using drills and curettes to more invasive open techniques. For example, premature closure of the distal radial physis can be associated with ulnar overgrowth, leading to altered wrist mechanics and pain. An epiphysiodesis of the ulna can prevent worsening of the deformity (Figure 8).

Scheider et al. [49] reported seven cases with the diagnosis of a painful ulnar positive variance in four individuals who underwent a temporary epiphysiodesis. This was done using a customized shortened 1.0 mm thick nonlocking two- or three-hole plate with 2.3 mm wide screw holes and screw lengths between 10 and 14 mm. The average age at implantation was 12.4 years and 14.7 years at explantation. The mean ulnar variance of +3.9 mm preoperatively was reduced to +0.1 mm, which led to satisfactory results in six out of seven cases. One case needed a secondary ulnar shortening osteotomy, which can be explained by having too little residual growth of the physis remaining at the beginning of therapy.

Campbell et al. [63] followed 31 wrists in 30 patients with premature distal radius physeal closure. Patients had an average age of 13.8 years [SD 1.6] at the time of surgery and were followed for a median of 163 days (ICR 101-419). The success rate of the procedure for the total group was 93.5%. However, because there were additional procedures performed at the time of epiphysiodesis in 67.7% of patients, including ulnar shortening osteotomies and distal radius osteotomies, the exact contribution of isolated epiphysiodeses could not be extracted from these results.

Waters et al. [36] followed thirty adolescents who underwent surgery after posttraumatic distal radial growth arrest at the average age of 14.8 years. Patients underwent ulnar epiphysiodesis in 11 cases and a combined radial and ulnar epiphysiodesis in three cases. Average ulnar variance among all patients improved from 4 mm positive (range −9 mm to +13 mm) before the procedure to 0 mm (range −6 mm to +4 mm) at the most recent follow-up radiographic evaluation (*p* < 0.01).

In a study by Farr et al. [40], performed on children with Madelung deformity, a series of 10 wrists out of 41 received an ulnar epiphysiodesis. Of these ten wrists, none of them required another intervention in correcting the deformity. The mean age of performed procedures was 13.4 ± 1.5 years. The authors postulate that ulnar epiphysiodesis may be considered in skeletally immature children older than 10 years of age with Madelung deformity.

### 6.4. Physeal Bar Resection

Resection of a physeal bar can be indicated in young children with a partial physeal closure, with the aim of restoring growth. The procedure for the removal of a physeal bar was first introduced by Langenskiöld [64] and is currently still being used in modified approaches. Success rates range from 15% to 38%, depending on the size and location of the bar [65]. Patients should have at least 50% of a healthy physeal surface in addition to 2 years of skeletal growth remaining [65,66]. Peterson et al. [67] classified the type and locations of a physeal bar into three subtypes: central, peripheral, and linear. A peripheral bar can be approached directly. Herein, excision of the overlying periosteum and removal of abnormal bone is carried out until the normal physeal cartilage is exposed completely. The remaining cavity is often interposed using fat or wax. Central and linear bars are more difficult to locate and visualize accurately. Preoperatively, the physeal bar must be identified correctly, preferably by computed tomography (CT) [65,68]. Fluoroscopy can be used to visualize the bar intraoperatively, but this may sometimes be difficult. In recent years, the use of a CT-guided navigation system helped identify the location, while an endoscope enables direct visualization of the physeal bar [66]. During follow-up, early magnetic resonance imaging (MRI) within four weeks has shown signs of incomplete resection [65].

### 6.5. Osteotomy

In severe deformities or in cases with too little growth remaining, a corrective osteotomy can be performed in addition to or without epiphysiodesis to correct the length and restore the anatomical alignment (Figure 9). In the forearm, performing a dome or wedge osteotomy allows for an accurate correction of alignment and restoration of the axial length, but it is an invasive procedure with a longer recovery time than epiphysiodesis. Patients with a cubitus varus may need a rotational correction in addition to angular correction. In these cases, either a dome osteotomy or a closed lateral wedge osteotomy is a reliable and powerful method to achieve correction [25]. In isolated growth arrest of the radius, an ulnar shortening osteotomy may be needed to correct the ulnar overgrowth [63].

## 7. Timing of Intervention

The timing of epiphysiodesis is crucial when planning guided correction of a limb or bone-length inequality, e.g., in the forearm. Estimating limb-length inequality starts with an estimation of the length of the unaffected limb at skeletal maturity. This is followed by determining the growth rate of the affected limb compared to the rate of the unaffected limb. The difference in rates can then be used to estimate the final limb-length inequality [69]. It is important to realize that not all length discrepancies increase continuously over time. Shapiro et al. described five different patterns of growth in lower-extremity length discrepancies [70]. A Type-I proportionate progression pattern was seen in children with destroyed physes. In this type, the length discrepancy develops and increases continually with time at the same proportionate rate. This allows for the estimation of the ultimate limb length or growth remaining.

A multitude of methods to accurately determine final limb length or remaining growth have been developed over the course of the years. Anderson and Green first introduced growth-remaining charts using skeletal age [60]. This method was later simplified by the introduction of the Moseley straight-line graphs in which only skeletal maturity is used [61]. The Rotterdam straight-line graph can be seen as an improvement of the Moseley straight-line graphs by means of a further expansion of that database [71]. The White–Menelaus formula uses chronological age and is based on a simple calculation with the assumption of a fixed mean annual growth and the assumption of physeal closure at a specific age for boys and girls [72]. The Paley multiplier method (MM) also uses chronological age and is based on an age-based multiplier specific for each age to calculate the final limb growth and remaining growth [73].

Each method has its pitfalls and potential advantages, but none of them is universally accepted as the gold standard in determining the timing of epiphysiodesis. For the upper limb, only the Anderson and Green-based growth charts of Stahl et al. and the Paley multiplier have been developed [74,75]. Birch and Makarov compared different methods of limb length prediction and found skeletal age to be superior to chronological age for prediction [76,77]. Sanders et al. [69] compared both the MM and skeletal age measurements and found that chronological age was superior to skeletal age for predicting ultimate limb length in children prior to their adolescent growth spurt. In contrast, after the start of the growth spurt, predicting limb length using skeletal age proved superior. This observation is consistent with other studies [60,78].

This raises further questions as to why chronological age is a better predictor prior to the adolescent growth spurt but worse after it. The Paley multiplier was based on the assumption of a Shapiro type-I linear growth pattern that remains the same during maturity and the multipliers remaining the same regardless of the growth phase the child is in. Differences in accuracy between the MM and the other methods might be due to each method having its own way of taking the growth spurt into account. Sanders et al. suggested that using peak height velocity (PHV) was the best marker for maturity during the transition into adolescence [69]. Growth measurements can then easily be calculated using PHV-derived multipliers. These multipliers are currently made using skeletal age, which is closely related to PHV during adolescence.

Different methods of measuring skeletal maturity are currently used. Historically, the Greulich and Pyle atlas for hand bone age is the most widely known and used [79]. The Sauvegrain method assesses skeletal age from elbow radiographs based on a 27-point scoring system [80]. This method uses four ossification centers of the elbow: the lateral condyle, trochlea, olecranon apophysis, and proximal radial epiphysis as landmarks. The scores of these structures are summed, and a graph is then used to determine the skeletal age. In contrast to the GP atlas, this method allows for the assessment of skeletal age in 6-month intervals during the phase of accelerating growth velocity, which makes it markedly suitable for the period of growth spurt.

When compared to the Greulich and Pyle atlas, the Sauvegrain method has been proven to be a more accurate method for the assessment of skeletal age during puberty, with the addition of having a high inter- and intra-observer reliability [81,82]. Furthermore, it is shown that the Sauvegrain score is a reliable marker for measuring PHV in children [83]. It should therefore prove suitable for predicting PHV-derived multipliers used for a more accurate prediction of growth in children during their growth spurt.

## 8. Conclusions and Recommendations

Because of the ability of children to correct osseous deformities during longitudinal growth, often, the deformity is corrected naturally. Their frequent relation with joints, however, may potentially have harmful consequences if discovered late or left untreated. Therefore, early clinical evaluation is often helpful in giving a quick indication, followed by additional radiologic evaluation for a more concise measure of the deformity. Current methods of measuring humeral and radial alignment prove sufficient for children, regardless of their age. However, in measuring ulnar variance, the use of age-specific methods such as the Hafner method and the method of perpendiculars may additionally improve accuracy.

If surgical intervention is necessary, deformity correction by means of guided growth poses an elegant and low-invasive option. Surgical treatment options include (hemi-) epiphysiodesis, physeal bar resection, and osteotomy, as well as combinations of techniques. The preferred treatment option is dependent on the location of the deformity, involvement of the physis, presence of a physeal bar, and predicted length inequality at skeletal maturity.

An accurate estimation of the limb or bone length at skeletal maturity is crucial for the correct timing of intervention. To date, the Paley multiplier method based on chronological age remains the most accurate method for calculating final and remaining limb growth in the upper extremity. Multiple studies found that the calculation of growth using chronological age is superior prior to the growth spurt. Skeletal age is found to be more accurate during the growth spurt. These calculations are generally performed using peak-height velocity. The Sauvegrain method of skeletal age assessment using elbow radiographs proves to be a more simple and more reliable method than the current widespread method of assessment using hand radiographs by Greulich and Pyle. The Sauvegrain method also proves to be a reliable marker for measuring peak-height velocity in children.

For children prior to their growth spurt, the Paley multiplier method remains the most accurate and simple method. In the absence of other validated methods, the Paley method can additionally be used for predicting growth during the growth spurt. However, because the Sauvegrain method has been proven suitable for measuring the peak-height velocity, further research should be carried out on calculating and validating specific PHV-based multipliers. Furthermore, the Paley multiplier still needs validation in the upper extremity. When both validation studies have been performed, a subsequent algorithm can be developed using the MM prior to the growth spurt and PHV-based multipliers based on the Sauvegrain method during the growth spurt. This combines the current two best methods and allows for a more accurate calculation of limb growth in the upper extremity. Because growth is not a two-dimensional progress, deformities often do not conform to a single plane during further growth. Therefore, apart from calculating the projected upper limb growth, an accurate assessment of which direction the deformity grows in should be carried out. Therefore, more research is needed for a better assessment and growth prediction of osseous deformities in 3D.

## Figures and Tables

**Figure 1 children-10-00195-f001:**
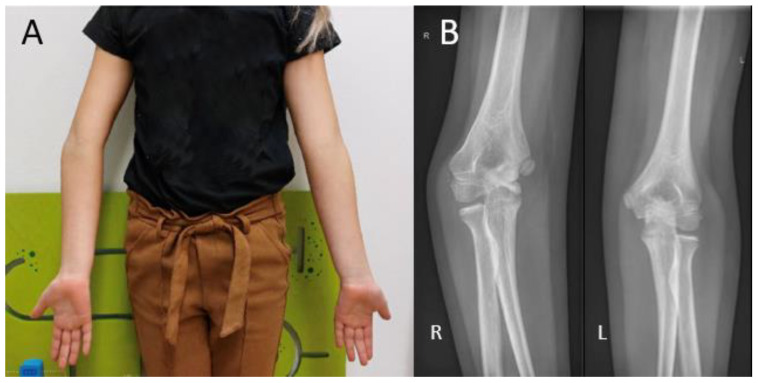
(**A**) Visual inspection of the carrying angle of the elbow in a 10-year-old girl showing a unilateral cubitus varus on the right side. (**B**) Anteroposterior radiographic views of the elbow with the unaffected contralateral side for comparison.

**Figure 2 children-10-00195-f002:**
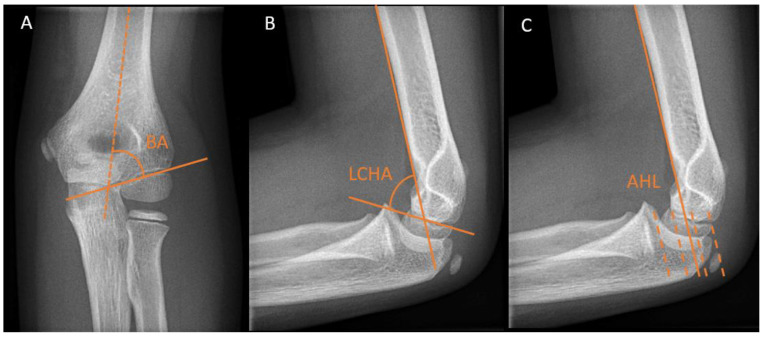
Distal humerus radiographic reference lines. (**A**) Baumann angle (BA) on an anteroposterior elbow view. (**B**) The lateral capitellohumeral angle (LCHA) on a lateral elbow view. (**C**) Anterior humeral line (AHL) on a lateral view. This line should pass between the two dotted lines in the middle.

**Figure 3 children-10-00195-f003:**
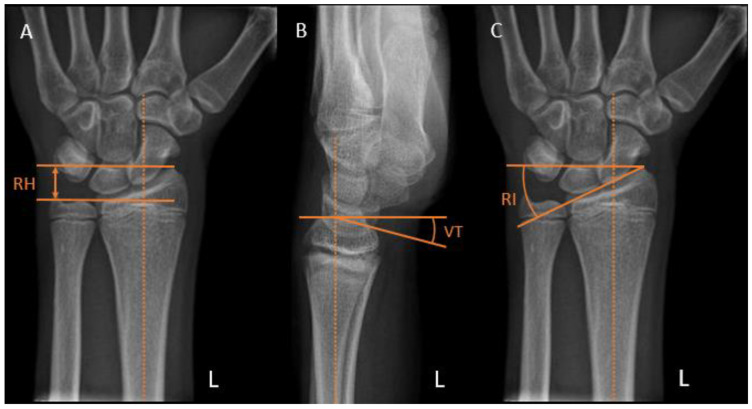
Distal radius radiographic reference lines. (**A**) Radial height (RH). (**B**) Volar tilt (VT). (**C**) Radial inclination (RI).

**Figure 4 children-10-00195-f004:**
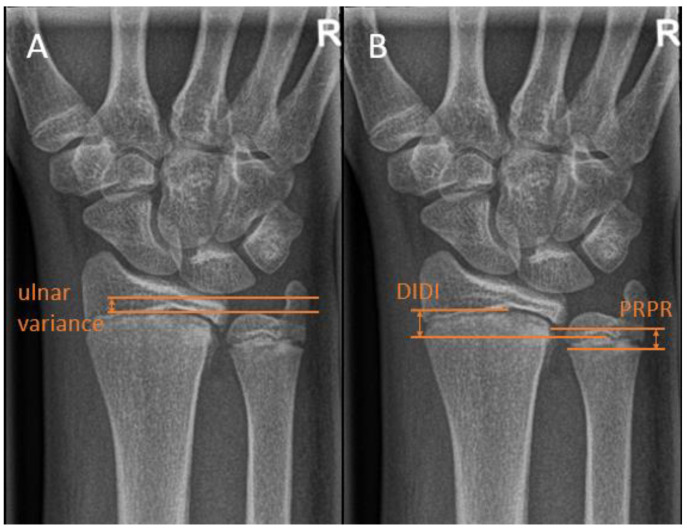
Distal ulna radiographic reference lines. (**A**) The Hafner method for measuring ulnar variance. (**B**) The method of perpendiculars for measuring ulnar variance, with ‘PRPR termed as the two most proximal points of the physis and ‘DIDI’ termed as the two most distal points of the physis.

**Figure 5 children-10-00195-f005:**
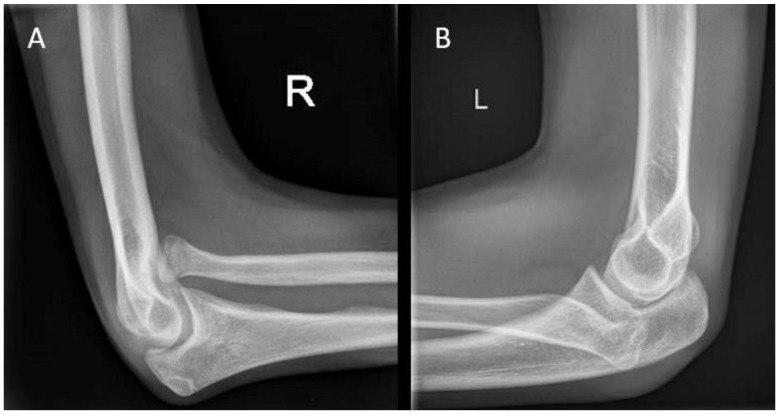
(**A**) A 17-year-old boy with congenital anterior radial head dislocations of the right arm. (**B**) A 16-year-old girl with congenital posterior radial head dislocations of the left arm, accompanied by a symptomatic elbow contracture. The girl was treated conservatively with a static progressive elbow flexion brace.

**Figure 6 children-10-00195-f006:**
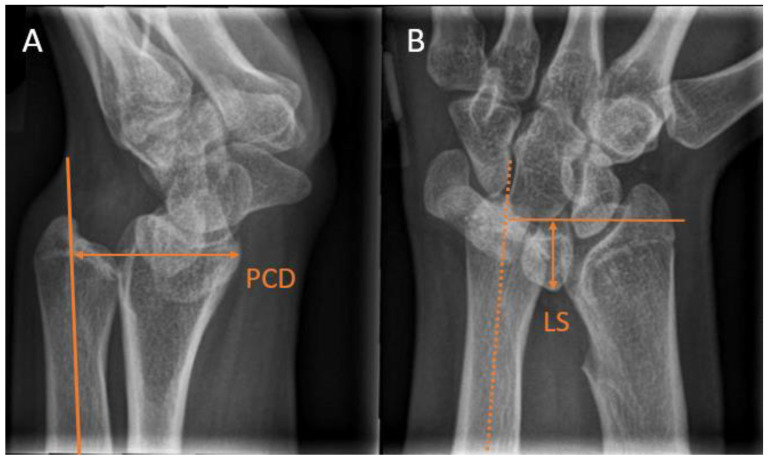
Radiographic wrist measurements used for the assessment of Madelung deformity. (**A**) Palmar carpal displacement (PCD) on a lateral wrist view, measured as the distance between the longitudinal ulna axis and the most volar lunate aspect. (**B**) Lunate subsidence (LS) on a posterioanterior view, measured as the distance between a perpendicular line to the longitudinal ulna axis and the most proximal lunate point.

**Figure 7 children-10-00195-f007:**
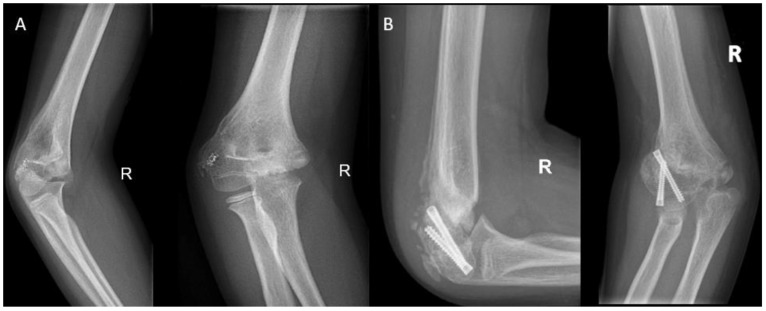
(**A**) Elbow radiographs of a 9-year-old girl with a posttraumatic cubitus varus, a flexion deficit of 60 degrees, and avascular necrosis of the medial condyle after a fall from height. (**B**) An epiphysiodesis using transphyseal screws was performed in addition to an arthrolysis with reduction of the coronoid fossa and release of the ulnar nerve.

**Figure 8 children-10-00195-f008:**
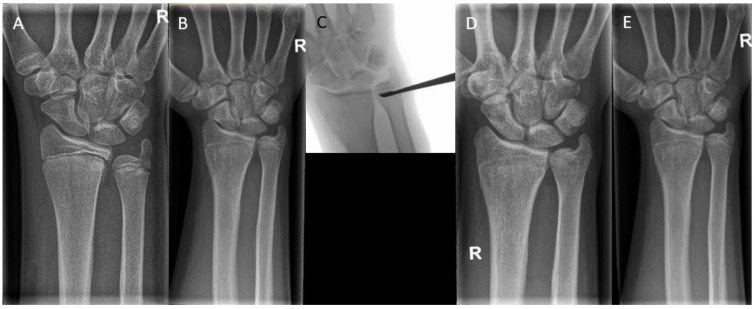
(**A**) A 15-year-old boy with premature closure of the distal radial physis after a Salter–Harris type 2/4 fracture. Initially, the boy had an ulna minus wrist. (**B**) A closed radial physis, accompanied by an impending ulna plus. (**C**) Intraoperative radiographs during epiphysiodesis of the ulna. (**D**) Postoperative radiographs show a closed physis of both the radius and the ulna. Note that the ulna had been growing until the epiphysiodesis, leading to an ulna zero. (**E**) Radiographs after 1-year follow-up. Note the unaltered ulnar variance.

**Figure 9 children-10-00195-f009:**
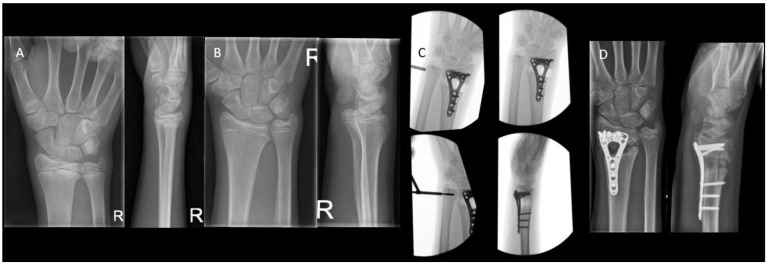
(**A**) A 17-year-old boy with a traumatic premature closure of the distal radial physis. (**B**) A closed radial physis, accompanied by an ulna plus. (**C**) Intraoperative radiographs during correction osteotomy of the radius combined with an epiphysiodesis of the ulna. (**D**) Radiographs six weeks postoperatively.

**Table 1 children-10-00195-t001:** Normal range of motion (ROM) of the elbow and wrist in degrees for both adults [4] and children [5].

Elbow	Adult ROM (Degrees)	Pediatric ROM (Degrees, SD)	Wrist	Adult ROM (Degrees)	Pediatric ROM (Degrees, SD)
Flexion	140	145 ± 5	Flexion	60	78 ± 6
Extension	0	1 ± 4	Extension	60	76 ± 6
Pronation	80	77 ± 5	Radial deviation	20	22 ± 4
Supination	80	83 ± 3	Ulnar deviation	30	37 ± 4

## Data Availability

Not applicable.

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
