# Peer review of "Indications and Timing of Guided Growth Techniques for Pediatric Upper Extremity Deformities: A Literature Review"

_children, 2023, doi:10.3390/children10020195_

Round 1
Reviewer 1 Report
The topic of the paper „Review Indications and timing of guided growth techniques for pediatric upper extremity deformities: a literature review” is very interesting for readers because osseous deformities in children arise due to progressive angular growth or complete physeal arrest and can be corrected using guided growth techniques; therefore, it is known very little about timing and techniques for the upper extremity.
This paper provides a review of the current literature on the clinical and radiological evaluation of normal upper extremity alignment and aims to provide state-of-the-art directions on deformity evaluation, treatment options, and optimal timing of these options during growth.
The authors concluded that more research is needed in the future for a better assessment and growth prediction of osseous deformities in 3D.
The introduction provides sufficient background and includes relevant references.
The manuscript is well written, and the text is easy to read.
The literature data are consistent and clearly presented.
The reference list is variously and relatively recently.
Author Response
"Please see the attachment."

Reviewer 2 Report
Dear Author,
Thank you for the opportunity to review this article.
It is an honest attempt to synthesize information about upper extremity deformities, that could be usefult in research and educational purposes.
It is a narrative review with zero patient pool, yet it is thorough and gathers information which needs an effort to combine. In line 233 you mention supracondylar fractures. Maybe it would be interesting to include fracture laterality in the discussion, regarding the functional aspect of deformities in everyday life, such as: The Relationship between the Dominant Hand and the Occurrence of the Supracondylar Humerus Fracture in Pediatric Orthopedics, published in Children, DOI 10.3390/children8010051.
It is a proper made manuscript and it is ready to being published after minor revision.
Author Response
"Please see the attachment."
